# A Study of Manufacturing Processes of Composite Form-Stable Phase Change Materials Based on Ca(NO_3_)_2_–NaNO_3_ and Expanded Graphite

**DOI:** 10.3390/ma13235368

**Published:** 2020-11-26

**Authors:** Yunxiu Ren, Chao Xu, Tieying Wang, Ziqian Tian, Zhirong Liao

**Affiliations:** Key Laboratory of Power Station Energy Transfer Conversion and System of Ministry of Education, School of Energy Power and Mechanical Engineering, North China Electric Power University, 2 Beinong Road, Changping District, Beijing 102206, China; 1152202045@ncepu.edu.cn (Y.R.); wty@ncepu.edu.cn (T.W.); 18401692885@163.com (Z.T.); zhirong.liao@ncepu.edu.cn (Z.L.)

**Keywords:** form-stable PCMs, operating parameters, optimization, thermal properties, microstructure

## Abstract

The fabrication of form-stable phase change materials (FS-PCMs) usually involves four manufacturing processes: mixing, immersion, stabilization, and sintering. In each process, the operation parameters could affect the performance of the fabricated PCM composite. To gain an efficient and low-cost method for large-scale production of the molten salts/expanded graphite (EG) composite FS-PCMs, the effects of different operating parameters were investigated, including the stirring speed, evaporation temperature, melt-impregnation, cold-pressing pressure, and sintering temperature on the densification, microstructure, and thermophysical properties of the composite FS-PCMs. It was found that the microstructure, the morphology and durability, and the thermophysical properties such as thermal conductivity and specific heat enthalpy depended highly on the operating parameters. The following optimal operating parameters of the Ca(NO_3_)_2_–NaNO_3_/EG composite FS-PCMs are suggested: the stirring speed of 20 rpm, the evaporation temperature of 98 °C, the melt-impregnation temperature of 280 °C, the cold-pressing pressure of 8 MPa, and the sintering temperature of 300 °C. The results of the present work can provide valuable insights for the large-scale production of the composite FS-PCMs.

## 1. Introduction

With growing interest in solar energy, off-peak electricity, human daily life and industrial waste heat, research works are being carried out for finding new phase change materials (PCMs) in the field of latent heat energy storage (LHES) systems. The LHES has the advantages of high storage density and melting temperature in the desired operational temperature range [1,2,3,4,5,6]. A good PCM should be abundant, available, and cheap to be widely used in applications, such as building [7,8,9,10], food packaging [11], underfloor heating system [12], air conditioning systems [13], and textile [14]. The feasibility of using a particular PCM for a specific application is based on some desirable thermal, physical, kinetic, chemical, and economical properties of the PCM, and the selection criteria for the various PCMs was reported by Elias et al. [15].

Compared with organic PCMs, inorganic PCMs [16] (e.g., molten salt) with large phase change temperature ranges and high latent heats have attracted more and more attention. However, the applications of molten salts in LHES systems are often restricted owing to their low thermal conductivity, liquid leakage, supercooling, phase segregation, and highly corrosivity [17]. Many researchers are exploring the solutions to solve the above-mentioned limitations of molten salts, such as encapsulation, direct incorporation, immersion, and form-stabilization of PCMs. Stabilization means the process of making something physically stable. During such a process, a large quantity of loose aggregate material is subjected to high pressure to cause the loose material to become a compact solid piece. Among them, form-stabilization for medium-high thermal applications (≥200 °C) has become a topic of interest in the past 20 years [18]. Form-stable PCM (FS-PCM) is typically achieved by incorporating PCM into porous supporting materials through capillary force and compacting the obtained powders into a geometric form [19,20]. The choice of supporting material for molten salts depends on many factors, including porosity, availability, thermal conductivity, and chemical compatibility with the PCMs. In previous studies, expanded graphite (EG), diatomite, vermiculite, multi-walled carbon nanotubes, perlite, kaolin etc., have been selected as the skeleton materials for PCMs, especially for medium-high temperature salts [21]. Among all these candidates, EG that has crevice-like and net-like pores, low density, high compressibility, high thermal conductivity, and low price is expected to be the effective reinforcement filler.

Recently, many investigations have been reported to develop desirable molten salts/EG FS-PCMs for practical engineering applications. Liu et al. [22] explored a novel process for preparing MgCl_2_-KCl eutectic salt/EG PCM blocks, which involved mixing solid molten salt with EG, compressing the mixture into a block, and then heating the block to a high temperature followed by cooling. Compared with the one prepared by the conventional method (first adsorption and then compression), the composite PCM blocks fabricated by the novel process exhibited better uniformity, excellent thermal reliability, and smaller volume expansion. It was found that the PCM blocks also exhibited a reduction in supercooling by 3.7 °C and an enhancement in thermal conductivity by 11-fold as compared with the eutectic salt. Sang et al. [23] fabricated the form-stable binary chlorides/EG composite materials by the melting impregnation method. With the EG content of 20 wt %, the FS-PCM showed a good thermal stability and a high thermal conductivity, which was 41.6 times higher than that of KCl-LiCl. The results showed that the KCl-LiCl was wrapped in the network structure of EG without agglomeration. Li et al. [24] found that both the 50 wt % LiNO_3_–45 wt % NaNO_3_–5 wt % KCl (LNK) ternary salts prepared by the static mixing melting method and the LNK/EG composite PCMs prepared by the ultrasound smashing method showed high phase change latent heats and chemical stability. With the addition of 15 wt % EG, the thermal conductivity of the composite PCM increased by 778% compared with LNK at the 20 MPa forming pressure. Most recently, we successfully developed a novel manufacturing method for the fabrication of Ca(NO_3_)_2_–NaNO_3_/EG composite FS-PCMs with a lower EG content, a high thermal conductivity, and a good thermal reliability [25].

From the literature review, the fabrication of FS-PCMs usually involves four manufacturing processes: mixing, immersion, stabilization, and sintering. In each process, the operation parameters (e.g., stirrer speed, evaporation temperature, melt-impregnation temperature, cold-pressing pressure, and sintering temperature) could affect the performance of the fabricated PCM composite. The effect of operating parameters is crucial for the optimization and fabrication of the FS-PCMs. However, no public report can be found about the effect of the operating parameters of the different fabrication processes on the microstructure, densification, and thermophysical properties of the composite FS-PCMs. Therefore, the aim of this work is to thoroughly investigate the effect of the operating parameters and establish a relationship between the thermophysical properties of the composite FS-PCMs and the operating parameters. In here, a eutectic Ca(NO_3_)_2_–NaNO_3_ salt mixture had been used as the PCM, and EG was selected as the supporting material. Through the specific tests, the influences of the operating parameters (including stirring speed, evaporation temperature, melt-impregnation, cold-pressing pressure, and sintering temperature) on the microstructure, densification, and thermal properties of the resulting materials were investigated, and the optimal operating parameters were obtained by the analysis of the results. The present study could provide deep insights about the influences of fabrication parameters on the performance of composite PCMs and benefit from the optimization of composite PCM fabrication.

## 2. Experimental Procedure

### 2.1. Raw Materials

#### 2.1.1. Supporting Materials

The melted PCMs can be easily retained within pores and adsorbed onto the flakes of EG, owing to their enormously high surface area-to-volume ratios. There are some desirable properties for choosing EG as the supporting material, just like chemical properties and economic aspects. Herein, EG was used as the supporting material for thermal conductivity enhancement. EG is a kind of modified graphite that has a layered structure with interlayer space [26]. It was prepared by immersing natural flake graphite into chromic acid and sulfuric acid, which forced the crystal lattice planes apart, thus expanding the graphite. Then, the mixture was dissolved in water, and the solution was filtered. The filtrate was collected and dried in the oven at temperature lower than 100 °C to ensure that the water did not boil. During this process, so-called “expandable graphite” was produced, as shown in Figure 1a. Finally, the expandable graphite was put into a vacuum muffle furnace with the expansion temperature starting at 800 to 1000 °C to expand with expansion ratios of 100, 150, 180, 200, 250, 280, and 300 times without oxidation [27,28]. The expanded graphite with an expansion ratio of 300 times is shown in Figure 1b. The raw materials of graphite flake (purity ≥99%) were obtained from Qingdao Co., Ltd., Qingdao, China

#### 2.1.2. PCM Materials

Calcium nitrate tetrahydrate (Ca(NO_3_)_2_·4H_2_O) (A.R. grade) and sodium nitrate (NaNO_3_) (A.R. grade) were purchased from Beijing Chemical Reagent Co., Ltd., Beijing, China. In our daily life, calcium nitrate (Ca(NO_3_)_2_) is commonly found as a calcium nitrate tetrahydrate (Ca(NO_3_)_2_·4H_2_O). The dehydration reaction of Ca(NO_3_)_2_·4H_2_O to anhydrous Ca(NO_3_)_2_ took place in two steps, in which all H_2_O molecules were liberated consecutively. After that, the binary eutectic Ca(NO_3_)_2_–NaNO_3_ was prepared by statically mixing calcium nitrate and sodium nitrate with the molar ratio of 3:7. More details of the eutectic salt synthesis were described previously [25]. Figure 2 illustrates the thermal characterization of Ca(NO_3_)_2_·4H_2_O, NaNO_3_ and the binary eutectic nitrate Ca(NO_3_)_2_–NaNO_3_/3–7 using the TG-DSC (differential scanning calorimeter, Netzsch STA449C, Selb, Germany) method. It was found from the TG-DSC curves that the binary eutectic nitrate melts at a single temperature that is lower than the melting points of the separate constituents of sodium nitrate or calcium nitrate.

### 2.2. Fabrication Methods of the Ca(NO_3_)_2_–NaNO_3_/EG Composite

#### 2.2.1. Melt-Impregnation Method

Melt-impregnation is a primary process for forcing the liquid salts to penetrate into the pores of the supporting materials. During this process, the supporting materials are immersed into the liquid PCMs, allowing for absorption by capillary action. Many previous studies have suggested that the inorganic PCM can be impregnated into EG by the conventional melt-impregnation method. For instance, Wu et al. [29] prepared the composite 50%Na_2_SO_4_·10H_2_O-50%Na_2_HPO_4_·12H_2_O/EG FS-PCMs by a physical blending and impregnation method. The hydrated salts were filled into pores or adhered onto the flakes of EG by physical interactions, including capillary forces and surface tension. Moreover, the composites fabricated by this method showed favorable thermal properties and low cost, making it promising for low-temperature thermal energy storage applications. Herein, we tried the simple melt-impregnation method by using EG (the mass fraction was 7 wt %) as the supporting material and the eutectic Ca(NO_3_)_2_–NaNO_3_ mixture as the PCM. The obtained Ca(NO_3_)_2_–NaNO_3_/EG mixtures were divided into four ceramic crucibles. Then, the four ceramic crucibles were placed in a vacuum furnace and maintained at 260, 280, 300, and 320 °C temperatures, respectively, for 5 h. However, it was verified that this is an ineffective method for forcing the liquid Ca(NO_3_)_2_–NaNO_3_ mixture into the pores of EG. The four different conditions yielded the same result in both morphology and structure. The digital photo and SEM image of the massive Ca(NO_3_)_2_–NaNO_3_/EG mixture heated by 280 °C are shown in Figure 3a,b. Notably, the capillary forces involved were not sufficient to allow liquid nitrate salts to pass through the surface of the porous materials. The result revealed that the conventional melt-impregnation method offered extremely low-permeability for the mixture with EG content of as low as 7 wt %. This could be caused by the fact that it was very difficult for EG to absorb a large quantity of liquid PCMs by simple immersion, owing to the pore space of the EG matrix being blocked up with the air [30]. Therefore, the conventional melt-impregnation method is ineffective for the fabrication of the Ca(NO_3_)_2_–NaNO_3_/EG mixture.

#### 2.2.2. Solution + Melt-Impregnation Method

To ensure that more nitrate salts permeate through the surrounding pores of EG, we tried a novel process, which is referred to as the “solution + melt-impregnation method”. First, the nitrate salts Ca(NO_3_)_2_ and NaNO_3_ with the total amount of 93 g were dissolved in 200 mL of deionized water to obtain a homogeneous solution, and the blending mole ratio of Ca(NO_3_)_2_/NaNO_3_ was fixed to be 7/3. Subsequently, EG with a fixed mass of 7 g was added into the solution and stirred vigorously in a magnetic stirrer at 58 °C for 10 min with a set stirring speed. Then, the mixture was dried in an oven at a set evaporation temperature for 10 h to remove moisture. The oven-dried Ca(NO_3_)_2_–NaNO_3_/EG composite PCMs were heated to 150 °C for 5 h, which ensured that the Ca(NO_3_)_2_·4H_2_O was evenly dehydrated into Ca(NO_3_)_2_. After that, the mixture was heated at 280 °C for 5 h in the vacuum muffle furnace until the EG was completely impregnated with the melted PCMs. The obtained Ca(NO_3_)_2_–NaNO_3_/EG composite is displayed in Figure 4. As shown in Figure 4a, no nitrate salts were observed on the surface of EG and the wall of the ceramic crucible. Based on Figure 4b, it can be seen that the salt particles were adsorbed and dispersed homogeneously into the porous network of EG. Therefore, it can be considered that the solution + melt-impregnation method solved the low-permeability problem (as shown in Figure 3a), and the nitrate salts were fully impregnated into the pores of EG or adhered to the surface of EG.

### 2.3. Manufacturing Processes for Producing the Form-Stable Composite

#### 2.3.1. Cold-Compressing Process

In this process, the mixed powders were uniaxially pressed into cylinders of 15 mm in diameter and 5 mm in length under a uniaxial pressure. To estimate the effect of cold-pressing pressure on the phase change behaviors of the Ca(NO_3_)_2_–NaNO_3_/EG composite FS-PCMs, different cold-pressing pressures (0, 4, 8, 12, and 16 MPa) were used in the present work, and the five samples were marked as R_1_, R_2_, R_3_, R_4_, and R_5_, respectively.

#### 2.3.2. Sintering Process

During the manufacturing process of the form-stable composite, the sintering process plays a key role. After stabilization, the obtained composite FS-PCMs did require a sintering process to control the densification and grain growth of the shaped materials. In the sintering process, the composite FS-PCMs were sintered in the vacuum muffle furnace at a high temperature for 5 h at the peak temperature with the heating rate of 10 °C/min, after which the samples were cooled down to the room temperature in the furnace. To investigate the effect of the sintering temperature on the properties of the composite, four different sintering temperatures (280, 300, 320, and 340 °C) were tested.

### 2.4. Characterization

Scanning electron microscopy (SEM) microanalysis were conducted on a TM-1000 (JSM-7800F Schottky, Tokyo, Japan). The hot disk thermal constants analyzer (Hot Disk, Hot Disk Co., Ltd., Uppsala, Sweden) was employed to measure the thermal conductivities of the samples at room temperature. The errors associated with this apparatus were found to be less than ±3%. The five samples with different cold-pressing pressures were analyzed by the differential scanning calorimeter (DSC, HSC-4, Beijing henven Co., Ltd., Beijing, China), and all experiments were carried out at an argon flow rate of 20 mL/min and a heating ramp of 10 °C/min in the aluminum crucible (25 µL; 22 mg aluminum). The specific heat capacities were measured by the simultaneous thermal analyzer (NETZSCH STA 449F5, Netzsch, Selb, Germany). The uncertainty of the measurements was within ±5%.

## 3. Results and Discussions

In this section, the effects of the operating parameters including the stirring speed, evaporation temperature, melt-impregnation temperature, cold-pressing pressure, and sintering temperature on the microstructure, densification, and thermophysical properties of the composite FS-PCMs were discussed.

### 3.1. Effect of Stirring Speed on the Ca(NO_3_)_2_–NaNO_3_/EG Slurry

Stirring speed is a critical process parameter in the composite process that not only determines the homogeneous distribution of EG in aqueous solution of salts but also influences the loose and porous vermicular structure of EG. Therefore, we performed several runs at different stirring speeds: 0, 20, 40, and 60 rpm, as shown in Figure 5. From Figure 5, the Ca(NO_3_)_2_–NaNO_3_/EG mixture fabricated by a 0 rpm stirring speed ran into the problem of getting layered, owing to the large differences in density between EG and nitrate salts. On the contrary, after stirring for 10 min at 58 °C, a 20 rpm stirring speed showed some good results in uniform distribution, and the microstructure of EG remained intact. However, 40 and 60 rpm stirring speeds can destroy the loose and porous vermicular structure of EG. The above results revealed that the stirring speed influenced the microstructure and the uniform distribution of EG in the slurry. At a lower stirring speed with 10 min stirring time, the clustering of EG was obvious. As the stirring speed increased, a better distribution of EG in the slurries was formed. However, the fine structure of EG was destroyed when the stirring speed was bigger than 20 rpm. From the above results, it can be concluded that EG can be dispersed uniformly in an aqueous solution of salts after 10 min stirring with the speed of 20 rpm.

### 3.2. Effect of Evaporation Temperature on the Mixture

As mentioned in previous works, liquid water contained within the slurry can be dried by slow evaporation or boiling, and the results are shown in Figure 6. Unfortunately, once vapor has been produced by boiling, the nitrate salts are liberated upon boiling, as shown in Figure 6. It should be noted that the inter wall of the beakers and the upper surface of EG were covered by the salt particles after heating at 120 and 150 °C for 5 h. This result indicates that the heating temperature should be lower than its boiling point. Therefore, the slurry was heated to 98 °C to ensure that the water did not overheat, and then the slurry was dried up by maintaining its temperature at 98 °C for 5 h in a vacuum muffle furnace. Benefiting from this, the uniform distribution of Ca(NO_3_)_2_–NaNO_3_/EG mixture was finally obtained. Therefore, the evaporation temperature is a key parameter that influences the drying behavior. Although high temperatures lead to a shorter drying time due to more rapid heat and moisture transfer, they also lead to the leakage of soluble salts carried out by the boiling bubbles in boiling liquid. In conclusion, a low-temperature drying process is recommended to ensure that the salt particles are deposited homogeneously on EG. Due to the advantage of a low-temperature drying process, the leakage problem resulting from the boiling salt solution can be avoided.

### 3.3. Effect of Melt-Impregnation Temperature on the Ca(NO_3_)_2_–NaNO_3_/EG Powders

In this section, different melt-impregnation temperatures were conducted to analyze the microstructure changes of the Ca(NO_3_)_2_–NaNO_3_/EG composites. The prepared composites were poured into the ceramic crucibles and then placed in a vacuum muffle furnace and maintained at different temperatures (260, 280, 300, and 320 °C) for 5 h. After that, different Ca(NO_3_)_2_–NaNO_3_/EG composite powders were obtained, and the corresponding SEM images of the four composites are shown in Figure 7a,b.

It is noticeable that when the melt-impregnation temperature was 260 °C, the solid salt particles melted into a mushy phase, which resulted in a lower impregnation, as displayed in Figure 7a. Compared with the 280 °C melt-impregnation temperature (referring to Figure 7b), there were still a large number of salt particles accumulated on the surface of the EG, which was caused by the mushy phase with high viscosity and low mobility. From Figure 7b, it can be found that the salts heated by 280 °C were evenly infused into the mesopores of EG matrices. Interestingly, as the melt-impregnation temperature was further increased to 300 and 320 °C, there were no significant changes in the Ca(NO_3_)_2_–NaNO_3_/EG composites (as shown in Figure 7c,d), indicating that melt-impregnation can be well achieved when the temperature is no less than 280 °C. However, it was found that a high melt-impregnation temperature of 300 or 320 °C could finally lead to evident liquid leakage and cracks of the prepared FS-PCMs. For these reasons, 280 °C was suggested for the melt-impregnation process.

### 3.4. Effect of Cold-Pressing Pressure on Phase Change Properties of the Composite FS-PCMs

As shown in Figure 8, the latent heat capacities of the composite FS-PCMs decreased dramatically as the cold-pressing pressures increased. Two sequential endothermic/endothermic peaks occurred in the DSC curves of the samples R_1_–R_4_. This result indicated that the formation of two sequential exothermic and endothermic peaks was related to the cold-pressing pressure. Corresponding to the different cold-pressing pressures, the melting temperature/solidification temperature ranges of samples R_x_ (x = 1, 2, 3, 4 and 5) were 216.7–234.1/210.1–188.6 °C, 210.1–254.7/236.9–189.2 °C, 209.8–259.6/244.2–184.2 °C, 207.1–271.7/257.7–185.3 °C, and 207.1–271.7/257.7–185.3 °C, respectively. Additionally, it should be noted that once the cold-pressing pressure increased, the melting temperature decreased, indicating that the compaction behavior could result in the heat transfer enhancement. Therefore, the cold-pressing pressure for the composite FS-PCMs had an obvious influence on the phase change behaviors. As similarly reported by Ren et al. [31], the appearance of two sequential phases of the samples indicated that cold-pressing had a negative effect on the co-crystal structure.

Figure 9 presents the SEM images of the Ca(NO_3_)_2_–NaNO_3_/EG composite FS-PCMs with the cold-pressing pressures of 0, 4, 8, 12, and 16 MPa. As shown in Figure 9a, the Ca(NO_3_)_2_–NaNO_3_/EG composite powers consisted of a large number of scattered particles. Contrarily, the particles interlocked and interacted with each other, forming networks in the continuous phase, for the composites after cold-pressing, as shown in Figure 9b,c. With the increase in the cold-pressing pressure, a closely packed linked structure was observed in Figure 9d,e. R_4_ and R_5_ showed a very similar micro-porous structure and morphology, indicating that further increasing the cold-pressing pressure to above 8 MPa cannot effectively change the microstructure.

On the other hand, the density variation of the composites with the cold-pressing pressure is given in Figure 9f. The composite density can be effectively influenced by the cold-pressing pressure when the pressure was below 8 MPa, while it can be hardly changed by the compression pressure when the pressure was above 8 MPa. This indicates that the densification behavior of the composite FS-PCMs has almost reached its limit when the cold-pressing pressure of 8 MPa was used.

### 3.5. Effect of Sintering Temperature on Stabilization of the Composite FS-PCMs

Sintering is usually used to enhance the composite’s shape stabilization against thermal cycling. To further study the effect of sintering temperature on the distribution and migration of salt particles in the composite FS-PCMs, the different sintering temperatures were set to 280, 300, 320, and 340 °C under the conditions of one sintering time of 5 h. The SEM images of them before and after sintering were shown in Figure 10a–e. It can be seen from Figure 10b,c that no leakage was observed from the surfaces after heating at 280 and 300 °C for 5 h, indicating that no liquid PCMs was released from the composite FS-PCMs during the heating process. In contrast, when the sintering temperature exceeded 320 °C, obvious cracks appeared on the surfaces of the composite FS-PCMs, as displaced in Figure 10d,e. Compared with other temperatures as mentioned above, the temperature of 300 °C was more suitable for sintering the composite FS-PCMs, and the relevant SEM images taken from the cross-sectional surface of the specimen R_3_ before and after the sintering at 300 °C are given in Figure 10a,c. Comparing Figure 10a with Figure 10c, it can be seen that after the sintering, a new bond was formed between the salt particles and the EG matrix. This was attributed to the fact that sintering at the temperature of 300 °C was sufficient to obtain the desired denser material, which consisted of abundant micro-pores to hold onto the liquid state PCMs. As a result, liquid salts filled the pores of the composite and diffused into the interlayer space of EG to form topological interlocking structures. From the above results, it was concluded that the sintering temperature has a strong influence on recrystallization. This statement is quite consistent with those of Erdogan et al. [32] and Falodun et al. [33], which reported that thermal softening could lead to smoothening of the material and grain reshaping.

### 3.6. Thermophysical Properties of the Composites

#### 3.6.1. Thermal Conductivity Improvement of the Composite FS-PCMs

Thermal conductivity affects the rate of energy storage and release, and it is one of the most important factors in TES applications. Thermal conductivities of the binary eutectic salts and the composite FS-PCMs were measured at 25 °C, and the results are shown in Figure 11. All samples were fabricated with the following optimal operating parameters: the stirring speed of 20 rpm, the evaporation temperature of 98 °C, the melt-impregnation temperature of 280 °C, and the sintering temperature of 300 °C. R_1_ without cold-pressing exhibited the lowest conductivity of 0.68 W/(mK). In general, the thermal conductivity of the composites can increase significantly after cold-pressing. With the increase of the cold-pressing pressure, the thermal conductivity of the composite firstly increased, reaching a maximum of 5.66 W/(mK) at 8 MPa, and then it began to decrease. This may be attributed to the surface cracks on the samples R_4_ and R_5_ formed by thermal stress during the sintering process, indicating that the interlocking structure could be destroyed when the cold-pressing pressure was too large [34]. This verified that the densification and microstructure of the composite FS-PCMs reached its optimal state under the cold-pressing pressure of 8 MPa.

#### 3.6.2. Specific Heat Capacities of the Composite FS-PCMs

The variation of specific heat capacity with the cold-pressing pressure of the Ca(NO_3_)_2_–NaNO_3_/EG composites at room temperature is given in Figure 12. Greater enhancement in the specific heat capacity (from 0.89 to 1.12 j/(kg·k)) was observed from the FS-PCM samples with the increasing cold-pressing pressure from 0 to 8 MPa. This was probably due to the increasing density of the FS-PCMs, which could result in rapid heat transfer. However, with the cold-pressing pressure increasing from 8 to 16 MPa, the corresponding specific heat capacity was kept nearly unchanged, which was consistent with the fact that the density of the composite remained unchanged when the pressure was larger than 8 MPa.

#### 3.6.3. Cycling Stability of the Composite FS-PCMs

The prepared composites R_4_ and R_5_ had serious problems such as thermal expanding, cracking, and leakage, as shown in the inset of Figure 10. The results show that the R_1_ and R_2_ had greater mass losses than the sample R_3_. Therefore, in the present study, the composite R_3_ was investigated only. In this section, only the sample R_3_ was tested due to its superiority compared with other samples. The cycling stability of sample R_3_ was evaluated by a heating–cooling cycle device, and the results are presented in Figure 13. It is seen that the enthalpy changes of the composite FS-PCM after 500 heating–cooling cycles were negligible. After 500 cycles, the enthalpies of melting and freezing were 84.5 kj/kg and 80.23 kj/kg [25], dropping by 5.89% and 8.48%, respectively. As presented in Figure 13b, both the melting and freezing temperatures changed slightly during the first 300 cycles, and they remained nearly unchanged in the last 200 cycles. As stated in our previous work [25], the prepared composite R_3_ showed a good thermal reliability in terms of minor changes in its thermal stability and phase change behaviors.

## 4. Conclusions

In this work, the effects of different fabrication parameters on the microstructure, densification, and thermal properties of the Ca(NO_3_)_2_–NaNO_3_/EG composite FS-PCMs were investigated. It was found that the optimal operating parameters of the Ca(NO_3_)_2_–NaNO_3_/EG composite FS-PCMs are the stirring speed of 20 rpm, the evaporation temperature of 98 °C, the melt-impregnation temperature of 280 °C, the cold-pressing pressure of 8 MPa, and the sintering temperature of 300 °C. The optimal operating parameters are very suitable for the large-scale production of the composite FS-PCMs. Additional conclusions can be drawn as below:(1)The layered problem resulting from the large differences in densities between EG and nitrate salts can be solved by the solution + low temperature drying technique. Compared with the melt-impregnation method, this method was more simple and effective.(2)With a temperature lower than 280 °C during the melt-impregnation process, the solid salt particles melted into a mushy phase, which resulted in a lower efficiency in impregnation, while a higher temperature was not beneficial for the forming and shaping processes. For these reasons, the 280 °C temperature is suggested for the melt-impregnation process.(3)Densification behavior was found to enhance the heat transfer of the composite FS-PCMs. However, the formation of the two sequential phases indicated that cold-compressing had a negative effect on the co-crystal structure.(4)The sintering process had a strong influence on recrystallization, and new bonding between the salt particles and EG matrices was formed.

## Figures and Tables

**Figure 1 materials-13-05368-f001:**
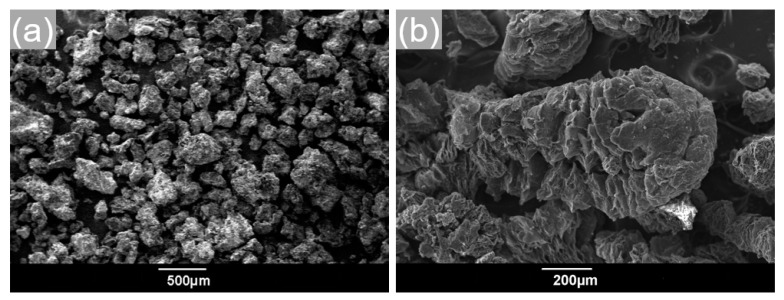
SEM images of (**a**) expandable graphite, and (**b**) expanded graphite.

**Figure 2 materials-13-05368-f002:**
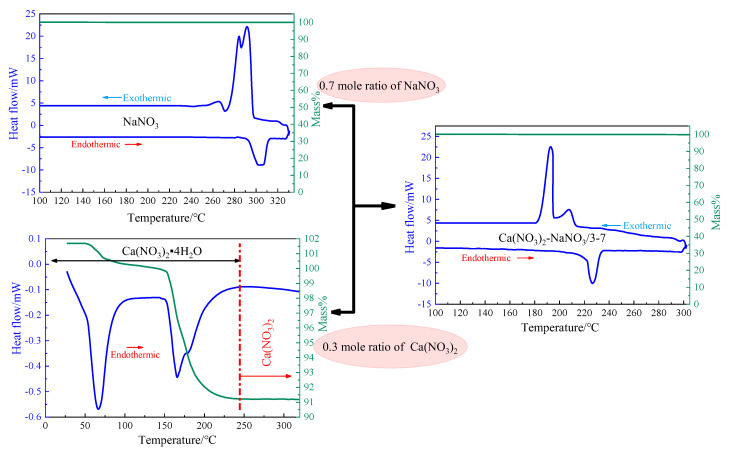
TG-DSC (differential scanning calorimeter) curves of sodium nitrate (NaNO_3_), calcium tetrahydrate nitrate (Ca(NO_3_)_2_·4H_2_O), and the binary eutectic nitrate (Ca(NO_3_)_2_–NaNO_3_/3-7).

**Figure 3 materials-13-05368-f003:**
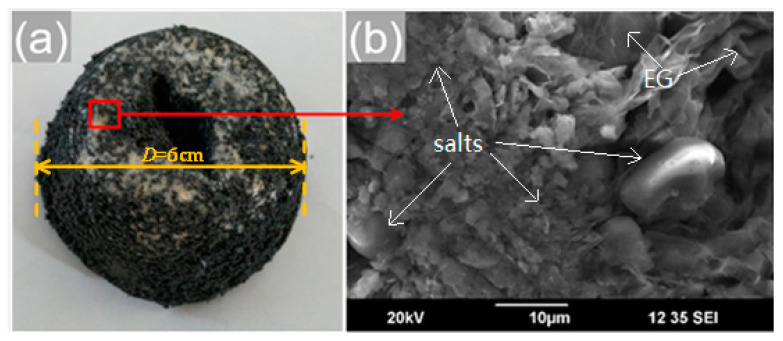
(**a**) Digital photo and (**b**) SEM image of the Ca(NO_3_)_2_–NaNO_3_/EG composite synthesized by the melt-impregnation method.

**Figure 4 materials-13-05368-f004:**
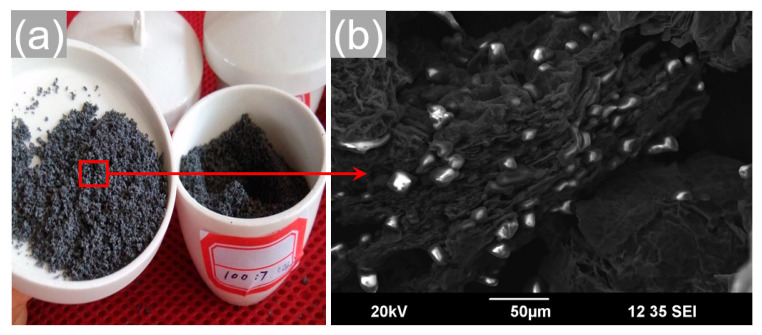
(**a**) Digital photo and (**b**) SEM image of the Ca(NO_3_)_2_–NaNO_3_/EG composite powders synthesized by the solution + melt-impregnation method.

**Figure 5 materials-13-05368-f005:**
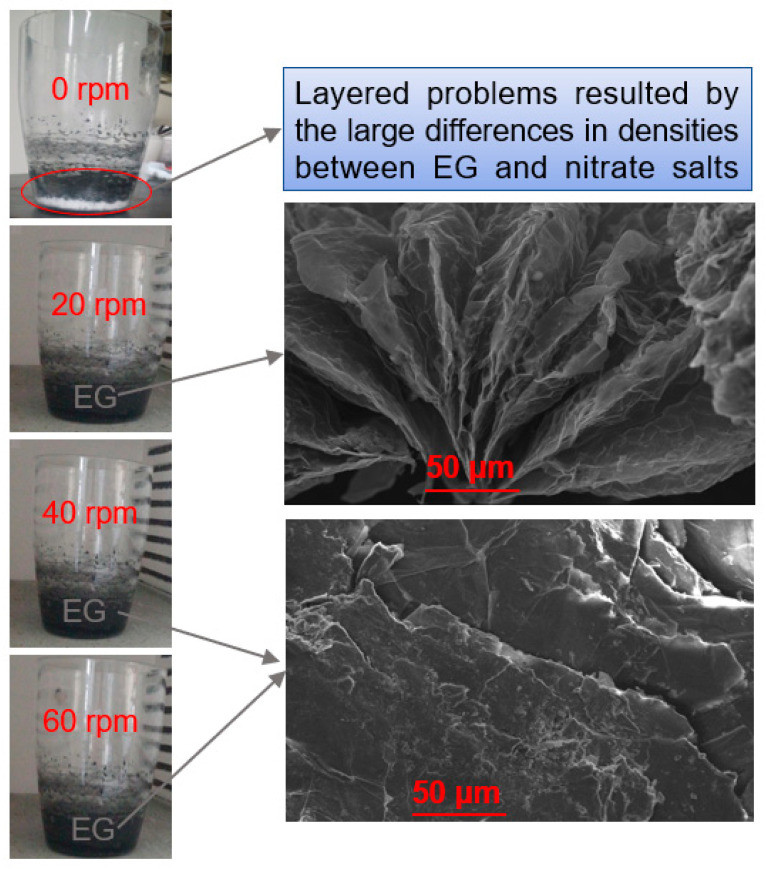
Photographs of the Ca(NO_3_)_2_–NaNO_3_/EG composite mixtures stirred by different stirring speeds: 0, 20, 40, and 60 rpm.

**Figure 6 materials-13-05368-f006:**
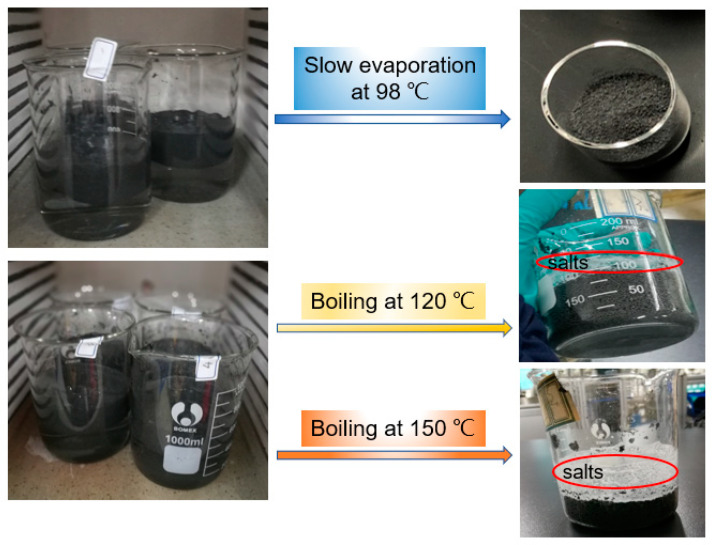
Photographs of the Ca(NO_3_)_2_–NaNO_3_/EG composite slurries dried at 98, 120, and 150 °C for 5 h.

**Figure 7 materials-13-05368-f007:**
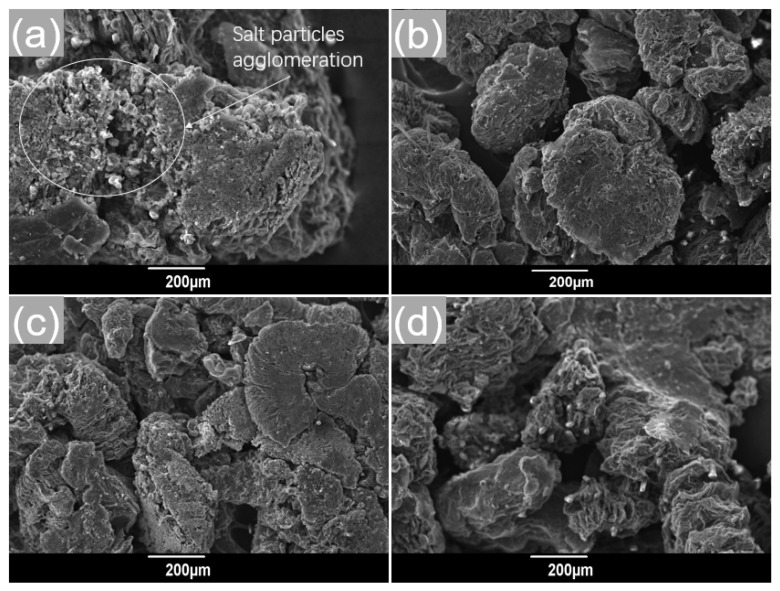
SEM images of the Ca(NO_3_)_2_–NaNO_3_/EG (expanded graphite) composite powders after heating at (**a**) 260 °C, (**b**) 280 °C, (**c**) 300 °C, and (**d**) 320 °C for 5 h.

**Figure 8 materials-13-05368-f008:**
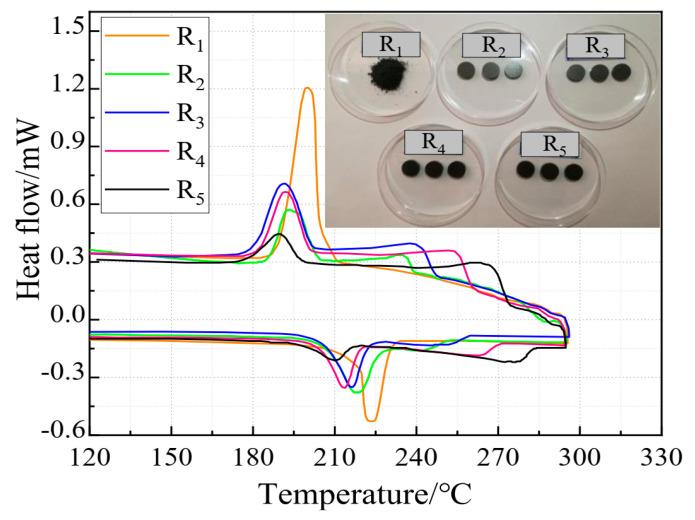
DSC curves of the five samples sintered at temperatures 120–300 °C.

**Figure 9 materials-13-05368-f009:**
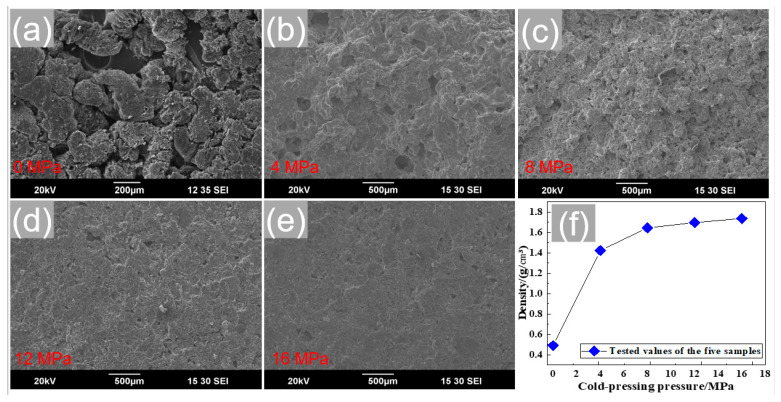
SEM images of the Ca(NO_3_)_2_–NaNO_3_/EG composites under different pressures: (**a**) 0 MPa, (**b**) 4 MPa, (**c**) 8 MPa, (**d**) 12 MPa, and (**e**) 16 MPa; and (**f**) effect of cold-pressing pressures on density of the composites.

**Figure 10 materials-13-05368-f010:**
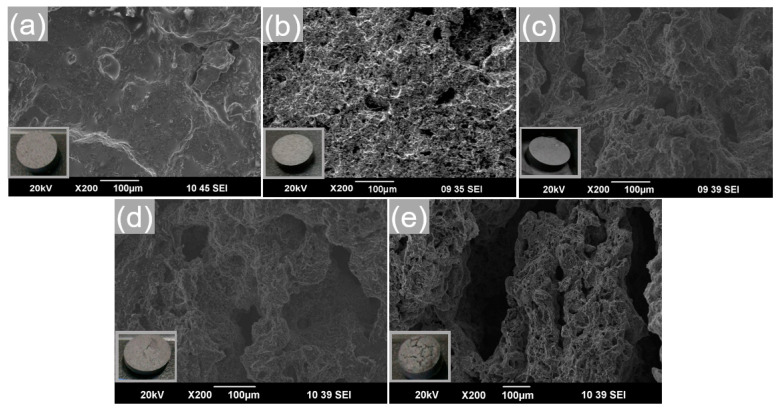
SEM images of the cross-sectional surfaces of the sample R_3_ before (**a**) sintering and after sintering at (**b**) 280 °C, (**c**) 300 °C, (**d**) 320 °C, and (**e**) 340 °C for 5 h.

**Figure 11 materials-13-05368-f011:**
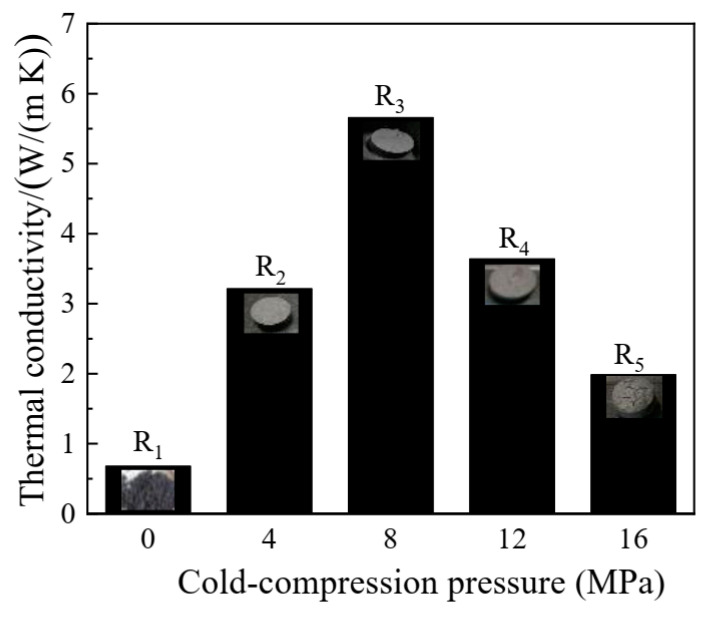
Thermal conductivities of the five samples after the sintering at 300 °C for 5 h.

**Figure 12 materials-13-05368-f012:**
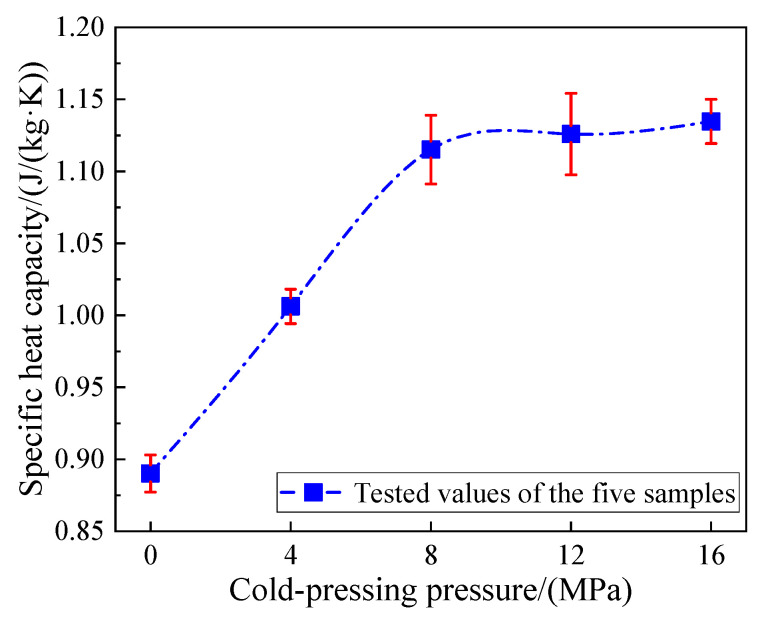
Variation of the specific heat capacity of the Ca(NO_3_)_2_–NaNO_3_/EG composite FS-PCM with the cold-pressing pressure.

**Figure 13 materials-13-05368-f013:**
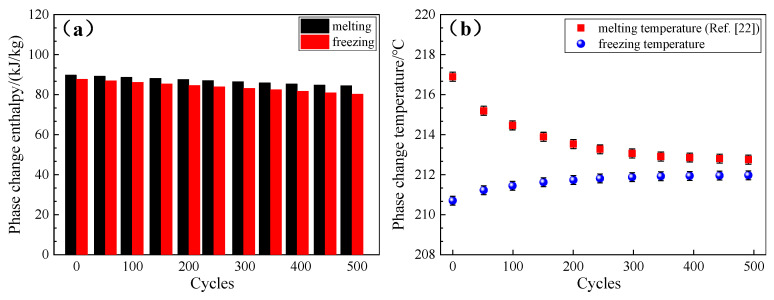
Thermal properties of the sample R_3_ after different cycles: (**a**) phase change enthalpy and (**b**) phase change temperature.

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
