# Peer review of "A Study of Manufacturing Processes of Composite Form-Stable Phase Change Materials Based on Ca(NO3)2–NaNO3 and Expanded Graphite"

_materials, 2020, doi:10.3390/ma13235368_

Round 1

Reviewer 1 Report

The authors presented the results about the different conditions of form-stable phase change materials based on Ca(NO3)2-NaNO3 and expanded graphite preparations to get the best composite. The work is interesting and clearly written, but it needs minor fixes.

  1. in the introduction the ref. 13 is missed.
  2. line 107- in what temperature was dried the solution. The wording" in low temperature" is not enough.
  3. I think that the authors should create a separate section called maybe "FS-PCM characterization methods" in which You will describe the methods line SEM, TG-DSC and DSC thanks to which You present the prepared composites and their thermal properties.
  4. in section 2.2. PCM materials, could You explain the TG-DSC results. Why after mixing Ca(NO3)2x4H2O and NaNO3 there is only one peak during the melting process? There is no information on what apparatus were the measurements done. 
  5. line 189- what conditions were used during the DSC analysis (see the comment no 3.)
  6. lines 203-206- I think it is not necessary to describe what will be presented in section "results and discussion". Please reject this paragraph.
  7. line 229-231- I think that all of us know what the boiling state is. Please reject this sentence.
  8.  line 275- I think it is not necessary to write the DSC conditions here (see comment no 3.)
  9. line 288- the wording "ref.[28]" should be convert to Ren et al [28], the same for ref. [29,30] in line 332.

Author Response

Response to Reviewer 1 Comments

The authors presented the results about the different conditions of form-stable phase change materials based on Ca(NO3)2-NaNO3 and expanded graphite preparations to get the best composite. The work is interesting and clearly written, but it needs minor fixes.

Point 1: In the introduction the ref. 13 is missed.

Response 1: Thank you for your comment. Ref. 13 is added.

Point 2: Line 107- in what temperature was dried the solution. The wording" in low temperature" is not enough.

Response 2: Following this suggestion, the "in low temperature" has been revised as follows:

“The filtrate was collected and dried in the oven at temperature lower than 100 oC to ensure the water didn’t boil.”

Point 3: I think that the authors should create a separate section called maybe "FS-PCM characterization methods" in which You will describe the methods line SEM, TG-DSC and DSC thanks to which You present the prepared composites and their thermal properties.

Response 3: Thank you for your comment and suggestions. In the revised paper, we added Section 2.4. as follows:

“Scanning electron microscopy (SEM) microanalysis were conducted on TM-1000 (JSM-7800F Schottky, Japan). The hot disk thermal constants analyzer (Hot Disk) was employed to measure the thermal conductivities of the samples at room temperature. The errors associated with this apparatus were found to be less than ±3%. The five samples with different cold-pressing pressures were analyzed by the differential scanning calorimeter (DSC, HSC-4), and all experiments were carried out at an argon flow rate of 20 ml/min and a heating ramp of 10 oC/min in the aluminum crucible (25 µL; 22 mg aluminum). The specific heat capacities were measured by the simultaneous thermal analyzer (NETZSCH STA 449F5). The uncertainty of the measurements was within ± 5%.”

Point 4: In section 2.2. PCM materials, could You explain the TG-DSC results. Why after mixing Ca(NO3)2x4H2O and NaNO3 there is only one peak during the melting

Response 4: Thank you for your comment. In fact, the binary eutectic Ca(NO3)2-NaNO3 was prepared by statically mixing Ca(NO3)2 (instead of Ca(NO3)2x4H2O ) and NaNO3 with the molar ratio of 3:7. And the eutectic Ca(NO3)2-NaNO3/3-7 is a homogeneous mixture of substances that melts or solidifies at a single temperature that is lower than the melting point of any of the constituents. Besides, we also added the following sentence in the revised paper:

“In our daily life, calcium nitrate (Ca(NO3)2) is commonly found as a calcium nitrate tetrahydrate (Ca(NO3)2·4H2O). The dehydration reaction of Ca(NO3)2·4H2O to anhydrous Ca(NO3)2 underwent in two steps, in which consecutively all H2O molecules were liberated. After that, the binary eutectic Ca(NO3)2-NaNO3 was prepared by statically mixing calcium nitrate and sodium nitrate with the molar ratio of 3:7.”

Point 5: Line 189- what conditions were used during the DSC analysis (see the comment no 3.)

Response 5: Thank you for your comment. The five samples with different cold-pressing pressures were determined by the differential scanning calorimeter (DSC, HSC-4), and all experiments were carried out at an argon flow rate of 20 ml/min and a heating ramp of 10 oC/min in the aluminum crucible (25 µL; 22 mg aluminum).

Point 6: Line 203-206- I think it is not necessary to describe what will be presented in section "results and discussion". Please reject this paragraph.

Response 6: Thank you for your comment and suggestion. Line 203-206 is the introduction of this section which could help the readers summarize the main focus of Section 3. Therefore, we think it is all right to keep the short sentences.

Point 7: Line 229-231- I think that all of us know what the boiling state is. Please reject this sentence.

Response 7: Based on this comment, this sentence has been deleted.

Point 8: Line 275- I think it is not necessary to write the DSC conditions here (see comment no 3.)

Response 8: Thank you for your comment. The corresponding sentences have been deleted.

Point 9: Line 288- the wording "ref. [28]" should be convert to Ren et al [28], the same for ref. [29,30] in line 332.

Response 9: Thank you for your comment and suggestion. The wording "ref. [28]" and "ref. [29,30]" had been convert to " Ren et al. [31]" and " Erdogan et al. [32] and Falodun et al. [33]", respectively.

We deeply appreciate the reviewer’s comments, which are very valuable for improving the quality of the present work.

Reviewer 2 Report

This is an interesting work on the interesting subject of the development of shape stabilised or encapsulated phase change materials.

Some points are identified below that will help authors to improve the manuscript.

Line 31-35. Authors refer to latent heat energy storage systems but the references 1-4 hardly describe such technology. Refs 1-3 describe PCM studies rather than LHES systems. The study of such LHES systems is described in the recent works such as:

Koukou et al. Experimental assessment of a full scale prototype thermal energy storage tank using paraffin for space heating application, International Journal of Thermofluids, Volumes 1–2, February 2020, 100003

Or

Du, J. Calautit, Z. Wang, Y. Wu, H. Liu,A Review of the applications of phase change materials in cooling, heating and power generation in different temperature ranges, Appl. Energy, 220 (2018), pp. 242-273

Line 35. The technical requirements of a suitable PCM for a specific applications are much more than the authors note. Hardly any PCMs meets all chemical, physical and economical requirements but since this journal is addressing materials issues please update the text by providing a list of selection criteria for the various PCMs or just for inorganic / salt hydrates. Authors may refer to Elias and Stathopoulos A comprehensive review of recent advances in materials aspects of phase change materials in thermal energy storage, Energy Procedia, 161 (2019) 385-394.

Line 45-47. I find the term “Form-stable PCMs” not very accurate for describing the resulting materials by mechanical mixing or absorption mixing of a PCM with a porous powder. The resulting substance is still a powder with no macroscopic geometrical shape or monolithic structure. I would suggest to avoid the term or provide clarifications.

Language needs polishing throughout the text: i.e. (line 99: …onto the pores…)

Line 106. It is not clear how flake graphite together with chromic acid and sulfuric acid are dissolved in water to form a solution that is filtered. After filtration was it the filtrate or the residue that was collected and dried top deliver the EG? Please clarify.

Line 121. Authors provide a binary eutectic nitrate without providing any information about its compositions – ratios. (This is only provided much later in the manuscript)

Fig 3a Please provide a scale

Fig3b the info is not clear at all. What is shown?

Line 165. What does 7% refers to? To suspension or to dissolved solids (nitrates)

What is the max uptake of such EG/PCM structures? Any leakage observed.

Line 172. EDs mapping is appropriate to provide evidence to the existence of PCMs

Line 185. The term “uniaxial” pressed is suggested instead of “static”

Line 194. What is the meaning of “stabilization” Please clarify

Line 197. What was the atmosphere during sintering/heat treatemtent?

Section 3.1 – What type of stirrer was used?

Fig 6. Please provide information about the composition of the residue on the beaker walls?

Line 265-266. At this point it is not clear how “it was found” that 300-320 oC heating could lead to liquid leakage and cracks of the as prepared FS-PCMs. Please explain.

Line 284-286. Authors assume that densification could enhance the heat transfer. However no density results are still shown. Please rephrase.

In Fig. 9. Please describe (a-f) in the caption.

Line 326. Please indicate :bonding” in the SEM picture (use arrows etc).

What is the phase change enthalpy of each sample? (R1-R5 and 280oC-340οC).

Line 347-351. How are these findings relate to the density results (Fig9f)?

Section 3.6.3 How is the PCM loading affected with  cycling? Is there any leaching or de-mixing observed?

Author Response

Response to Reviewer 2 Comments

This is an interesting work on the interesting subject of the development of shape stabilised or encapsulated phase change materials. Some points are identified below that will help authors to improve the manuscript.

Point 1: Line 31-35. Authors refer to latent heat energy storage systems but the references 1-4 hardly describe such technology. Refs 1-3 describe PCM studies rather than LHES systems. The study of such LHES systems is described in the recent works such as:

Koukou et al. Experimental assessment of a full scale prototype thermal energy storage tank using paraffin for space heating application, International Journal of Thermofluids, Volumes 1–2, February 2020, 100003 Or

Du, J. Calautit, Z. Wang, Y. Wu, H. Liu,A Review of the applications of phase change materials in cooling, heating and power generation in different temperature ranges, Appl. Energy, 220 (2018), pp. 242-273

Response 1: Based on this comment, we have added new references that are related to LHES systems to increase the adequacy of the literature review. The added references are listed as follows:

[5] M. Koukou, G. Dogkas, M. Vrachopoulos, J. Konstantaras, C. Pagkalos, V. Stathopoulos, P. Pandis, K. Lymperis, L. Coelho, A. Rebola, Experimental assessment of a full scale prototype thermal energy storage tank using paraffin for space heating application, Int. J. Heat Mass Transf. 1 (2020) 100003, https://doi.org/10.1016/j.ijft.2019.100003.

[6] K. Du, J. Calautit, Z. Wang, Y. Wu, H. Liu, Du, A review of the applications of phase change materials in cooling, heating and power generation in different temperature ranges, Appl. Energy 220 (2018) 242-273, https://doi.org/10.1016/j.apenergy.2018.03.005.

Point 2: Line 35. The technical requirements of a suitable PCM for a specific applications are much more than the authors note. Hardly any PCMs meets all chemical, physical and economical requirements but since this journal is addressing materials issues please update the text by providing a list of selection criteria for the various PCMs or just for inorganic / salt hydrates. Authors may refer to Elias and Stathopoulos A comprehensive review of recent advances in materials aspects of phase change materials in thermal energy storage, Energy Procedia, 161 (2019) 385-394.

Response 2: Based on this comment, we have added some sentences as marked red in the manuscript. Some of the added sentences are as follows:

“Besides, the feasibility of using a particular PCM for a specific application is based on some desirable thermal, physical, kinetic, chemical and economical properties the PCM, and the selection criteria for the various PCMs was reported by Elias et al. [15].”

[15] C. Elias, V. Stathopoulos, A comprehensive review of recent advances in materials aspects of phase change materials in thermal energy storage, Energy Procedia, 161 (2019) 385-394, https://doi.org/10.1016/j.egypro.2019.02.101.

Point 3: Line 45-47. I find the term “Form-stable PCMs” not very accurate for describing the resulting materials by mechanical mixing or absorption mixing of a PCM with a porous powder. The resulting substance is still a powder with no macroscopic geometrical shape or monolithic structure. I would suggest to avoid the term or provide clarifications.

Response 3: Following this comment, the “ Form-stable PCM (FS-PCM) is typically achieved by incorporating PCM into porous supporting materials through capillary force [19, 20]” has been changed to “ Form-stable PCM (FS-PCM) is typically achieved by incorporating PCM into porous supporting materials through capillary force, and compacting the obtained powders into a geometric form [19, 20].” in the revised manuscript.

Point 4: Language needs polishing throughout the text: i.e. (line 99: …onto the pores…)

Response 4: The writing of the manuscript has been carefully checked and revised.

Point 5: Line 106. It is not clear how flake graphite together with chromic acid and sulfuric acid are dissolved in water to form a solution that is filtered. After filtration was it the filtrate or the residue that was collected and dried top deliver the EG? Please clarify.

Response 5: Thank you for your comment. It was prepared by immersing natural flake graphite into chromic acid and sulfuric acid, which forced the crystal lattice planes apart, thus expanding the graphite. Then, the mixture was dissolved in water, and the solution was filtered. The filtrate was collected and dried in the oven at low temperature. During this process, a so-called “expandable graphite” was produced.

Point 6: Line 121. Authors provide a binary eutectic nitrate without providing any information about its compositions – ratios. (This is only provided much later in the manuscript)

Response 6: Thank you for your comment and suggestion. As mentioned in the sentence “binary eutectic nitrate Ca(NO3)2-NaNO3/3-7” in the manuscript, “3-7” is the molar ratio of Ca(NO3)2-NaNO3.

Point 7: Fig 3a Please provide a scale

Response 7: Thank you for your comment. Fig. 3a has been revised as follows:

Point 8: Fig3b the info is not clear at all. What is shown?

Response 8: Thank you for your comment. The enlarged area for Fig. 3a has been revised in Fig. 3b.

Point 9: Line 165. What does 7% refers to? To suspension or to dissolved solids (nitrates)

Response 9: Based on this comment, the sentence “EG with a fixed mass ratio of 7 wt.%” of present manuscript was changed into “EG with a fixed mass of 7 g.”

Point 10: What is the max uptake of such EG/PCM structures? Any leakage observed.

Response 10: Such a structure gave a high PCM mass fraction with 93 wt.%, and no leakage was observed.

Point 11: Line 172. EDs mapping is appropriate to provide evidence to the existence of PCMs

Response 11: Thank you for your comment and suggestion. EDs mapping is appropriate to provide evidence to the existence of PCMs, and more details were reported by our previous work as follows:

[34] Y. Ren, C. Xu, F. Ye, Z. Liao, The effect of the cold compressing pressure on the microstructure and thermal properties of binary eutectic nitrate/expanded graphite phase change material composites, Energy Procedia 158 (2019): 4598-4603, https://doi.org/10.1016/j.egypro.2019.01.748.

Point 12: Line 185. The term “uniaxial” pressed is suggested instead of “static”

Response 12: Following this comment, the "static" has been changed to "uniaxial" in the revised manuscript.

Point 13: Line 194. What is the meaning of “stabilization” Please clarify

Response 13: Thank you for your comment. “Stabilization” means the process of making something physically stable. During such a process, a large quantity of loose aggregate material is subjected to a sufficiently high pressure to cause the loose material to become a compact solid piece.

Point 14: Line 197. What was the atmosphere during sintering/heat treatemtent?

Response 14: During the sintering/heat treatment, the samples were placed in vacuum furnace.

Point 15: Section 3.1 – What type of stirrer was used?

Response 15: The suspension was stirred with a magnetic stirrer which consists of a water bath with magnetic stirring.

Point 16: Fig 6. Please provide information about the composition of the residue on the beaker walls?

Response 16: Thank you for your comment. It should be noted that the inter wall of the beakers and the upper surface of EG were covered by the salt particles, and the mole ratios of Ca(NO3)2 and NaNO3 of the residue on the beaker walls were 0.3 and 0.7 respectively.

Point 17: Line 265-266. At this point it is not clear how “it was found” that 300-320 oC heating could lead to liquid leakage and cracks of the as prepared FS-PCMs. Please explain.

Response 17: Thank you for your comment. The prepared FS-PCMs with the melt-impregnation temperature 280 and 320 oC had serious problems such as thermal expanding, cracking and leakage, as shown in the following Figure 1 (not shown in the paper). Therefore, 280 oC was suggested for the melt-impregnation process.

Figure 1. Photography of the prepared FS-PCMs with the melt-impregnation temperature from 280 to 320 oC.

Point 18: Line 284-286. Authors assume that densification could enhance the heat transfer. However no density results are still shown. Please rephrase.

Response 18: Thank you for your comment and suggestion. The “densification” in Line 284-286 are changed into “compaction”.

Point 19: In Fig. 9. Please describe (a-f) in the caption.

Response 19: The caption of Fig. 9 has been revised as follows:

Fig. 9. SEM images of the Ca(NO3)2-NaNO3/EG composites under different pressures: (a) 0 MPa, (b) 4 MPa, (c) 8 MPa, (d) 12 MPa, (e) 16 MPa; and (f) effect of cold-pressing pressures on density of the composites.

Point 20: Line 326. Please indicate: bonding” in the SEM picture (use arrows etc).

Response 20: Thank you for your comment and suggestion. As discussed in the corresponding section of the present work, “Comparing Fig. 10a with c, it can be seen that after the sintering, a new bond was formed between the salt particles and the EG matrix… As a result, liquid salts filled the pores of the composite and diffused into the interlayer space of EG to form topological interlocking structures.” Therefore, it’s difficult to assign arrows in the SEM picture for exhibiting the network topological interlocking structures.

Point 21: What is the phase change enthalpy of each sample? (R1-R5 and 280 oC-340 οC).

Response 21: Thank you for your comment and suggestion. We added the following sentence in the revised paper:

“The prepared composites R4 and R5 had serious problems such as thermal expanding, cracking and leakage, as shown in the inset of Fig. 10. Besides, the results show that the R1 and R2 had greater mass losses than the sample R3. Therefore, in the present study, the composite R3 were investigated only.”

Point 22: Line 347-351. How are these findings relate to the density results (Fig9f)?

Response 22: Thank you for your comment. In general, both the density and thermal conductivity of the composites can increase significantly after the cold-pressing.

Point 23: Section 3.6.3 How is the PCM loading affected with cycling? Is there any leaching or de-mixing observed?

Response 23: Thank you for your comment. As presented in Fig. 13a, the enthalpy changes of the composite FS-PCM after 500 heating-cooling cycles were negligible. Meanwhile, both the melting and freezing temperatures changed slightly during the first 300 cycles, and they kept nearly unchanged in the last 200 cycles, as depicted in Fig. 13b. The results indicate that the prepared composite R3 showed a good thermal reliability in terms of minor changes in its thermal stability and phase change behaviors. Therefore, PCM loading was not much affected with cycling, and no leaching or de-mixing was observed.

We deeply appreciate the reviewer’s comments, which are very valuable for improving the quality of the present work.

Round 2

Reviewer 2 Report

manuscript is improved